# The Prevalence and Genetic Diversity of *Babesia divergens* in *Ixodes ricinus* Nymphs Collected from Farm- and Woodland Sites in Ireland

**DOI:** 10.3390/pathogens11030312

**Published:** 2022-03-02

**Authors:** Fiona McKiernan, Amie Flattery, John Browne, Jeremy Gray, Taher Zaid, Jack O’Connor, Annetta Zintl

**Affiliations:** 1UCD Veterinary Sciences Centre, University College Dublin, D04 V1W8 Dublin, Ireland; fiona.mc-kiernan@ucdconnect.ie (F.M.); amie.flattery@ucd.ie (A.F.); taher.zaid@ucdconnect.ie (T.Z.); 2UCD School of Agriculture and Food Science, University College Dublin, D04 V1W8 Dublin, Ireland; john.a.browne@ucd.ie; 3UCD School of Biology and Environmental Science, University College Dublin, D04 V1W8 Dublin, Ireland; jeremy.gray@ucd.ie; 4MSD Ireland, South County Business Park, D18 X5R3 Dublin, Ireland; jack.oconnor@merck.com

**Keywords:** redwater fever, *Babesia divergens*, *Ixodes ricinus*, Ireland, tick infection rates, 18S rRNA

## Abstract

The parasite, *Babesia divergens* causes redwater fever in cattle and a rare, albeit life-threatening disease in humans. In Ireland, *B. divergens* has always been considered an important pathogen as the high incidence of redwater fever precluded areas of the country from cattle farming. Moreover a relatively large proportion of human cases were reported here. Red deer (*Cervus elaphus*), which often harbour babesias that are genetically very similar (if not identical) to *B. divergens,* are quite widespread. In this study 1369 nymphal *Ixodes ricinus* ticks collected from various habitats were screened for the presence of *B. divergens* using TaqMan followed by conventional nested PCR. Fragments of the 18S rRNA gene locus (560 bp) were compared against published Irish *B. divergens* isolates from cattle, humans and red deer. Overall just 1% of *I. ricinus* nymphs were infected with *B. divergens*, with similar infection rates in ticks collected from farm- and woodland. Most (90%) 18S rRNA gene fragments derived from woodland ticks were 100% identical to published sequences from cattle and humans. One differed by a single nucleotide polymorphism (SNP) as did two isolates from ticks collected in bogland. Two isolates derived from nymphs collected in farmland differed by 2 and 4 SNPs respectively.

## 1. Introduction

The protozoan parasite, *Babesia divergens,* is the causative agent of redwater fever, an important disease of cattle. Following transmission by hard 3-host tick, *Ixodes ricinus*, *B. divergens* invades the red blood cells of it host causing anaemia, haemoglobinuria and, in severe cases, anoxia and death [1]. The parasite also infects humans where it can cause acute, malaria-like disease. Though very rare (to date only about 60 cases have been reported in total since it was first described in the 1950s), and largely restricted to asplenic individuals, these zoonotic infections are usually life-threatening unless treated rapidly and aggressively [2].

The role of deer as potential reservoir hosts for *B. divergens* has long been the subject of debate. *Babesia* isolates that resemble *B. divergens* closely in the 18S rRNA gene region have been reported from a range of deer species, however whether these are identical to *B. divergens*, different strains or different species remains unclear. The only deer species where 18S rRNA and cytochrome oxidase subunit 1 (COI) gene fragments that were 100% identical to *B. divergens* have been reported are red deer [3,4], indicating that they are susceptible to *B. divergens* infection. This also makes them the most likely candidate for a wildlife reservoir host, although the extent to which they contribute to tick infection rates (if at all) is unknown.

In Ireland, *B. divergens* has always had a particular significance. The mild, humid climate is conducive to the tick vector and *I. ricinus* is widespread, especially along the western seaboard and the Shannon River system [5]. Up until the early 1990′s significant parts of the country were precluded from cattle farming because of the widespread occurrence of redwater fever. In fact the impact on the Irish cattle industry was such that a live pilot vaccine was developed in 1995 and used successfully immunise approx. 14,000 cattle in local trials [6]. While the incidence of redwater fever has declined in recent decades [7], pockets of high infection pressure remain where cases of acute disease are sporadically observed, particularly among bought-in replacement stock. With six out of the total of 60 human *B. divergens* cases reported here, Ireland also has had a comparatively large share of zoonotic *B. divergens* infections [8]. As a matter of fact the first patients to survive zoonotic infections were treated with imidocarb diproprionate under special license in University College Hospital, Galway [9]. Regarding *Babesia* infections in Irish deer, previous work has identified isolates from red deer that were 100% identical to bovine *B. divergens* sequences in two fragments of the 18SrRNA gene [3]. It is important to note at this point that the species composition of the Irish deer population is somewhat different from that of the rest of Europe in that roe deer, which are frequently infected with *Babesia* species other than *B. divergens*, are absent. On the other hand, red deer are relatively widespread. The other deer species present on the island are fallow deer (*Dama dama*), sika deer (*Cervus nippon*) and red-sika hybrids [10]. Overall, there is a comparatively good record of Irish *B. divergens* isolates from cattle, humans and deer available in GenBank [3,4,8].

This study aimed to investigate the potential role of deer in the transmission cycle of *B. divergens* in Ireland by (i) determining *B. divergens* infection rates in ticks collected in farmland, woodland and some bog and limestone pavement sites and (ii) by comparing the genetic diversity of isolates from ticks to published sequences from cattle, humans and red deer in Ireland.

## 2. Results

### 2.1. Tick Nymph Infection Rates with B. divergens

0.3% of the ticks collected from farmland and 1.6% of the ticks collected from woodland were found to be infected with *B. divergens* (Table 1). While none of the ticks collected from the Burren limestone pavement tested positive for *B. divergens*, the infection rate in nymphs collected in the bogland site was 4.1%. Due to the small number of infected ticks in all of the habitats, none of the differences were statistically significant.

### 2.2. Characterisation of B. divergens Isolates

The ~560 bp 18S rRNA gene fragments of 9 isolates derived from ticks collected in woodland were 100% identical to U16370 which is widely used as a reference sequence for 18S rRNA gene fragments of *B. divergens*. Three further isolates, including one from woodland and two isolated from ticks collected in bogland, differed from U16370 in a single base in position 640 but matched two published Irish bovine isolates (LC477140 and LC477142) in this position (Figure 1). The remaining two sequences (both from farmland) differed from U16370 by 2 and 4 SNPs respectively. One of these also had the SNP in position 640. These last two sequences had no 100% match in the database and were logged under accession numbers OL504563 and OL504564. 

## 3. Discussion

Considering that engorged ticks do not move very far and probably quest close to where they have dropped off [12], it is likely that the nymphs collected in woodland habitats had fed on wildlife hosts as larvae. Moreover, a significant proportion of them probably fed on deer, as these are the preferred hosts of all tick life cycle stages [13]. Interestingly, the *B. divergens* infection rates were similar in nymphs collected from woodland and farmland sites suggesting that woodland nymphs may have become infected by engorging on deer and indicating a potential role of (red) deer as reservoir hosts for *B. divergens*. However, it is important to stress, that due to the fragmented nature of the Irish landscape, deer are never far from farmland, just as cattle are never far from woodland.

In this study *B. divergens*-infected ticks were identified using a TaqMan PCR specific for the *B. divergens* heat shock protein 70, which, in the case of the canine *Babesia* complex was found to be more discriminatory than the 18S rRNA gene [14]. It is reasonable to assume that the same is true for other *Babesia* species. However, since the only deer *Babesia* hsp70 gene sequences currently available in the database are for *Babesia odocoilei*, the possibility that the PCR developed by Michelet and colleagues [15] also amplifies other, closely related, wildlife species cannot be excluded categorically. Nevertheless it is important to note that the primers and probes used in the protocol do not match the hsp70 gene of *B. odocoilei*.

Of course, it is now also well established that identification of *B. divergens* based on fragments of the 18S rRNA gene is also problematic, as many published bovine, human and red deer *B. divergens* isolates differ from the ‘reference sequence’ U16370 in one or two SNPs over the length of the whole gene (Figure 1). While the majority of our tick isolates matched U16370 by 100%, other sequences differed by 1 to 4 SNPs, which, considering the shortness of the fragment, was significant. Interestingly one of these SNPs, in position 640, had also been recorded in two published Irish bovine isolates. It was also surprising that the two isolates from nymphs collected in farmland were more heterogenous than those collected from woodland. As *B. divergens* is a major pathogen of cattle, one would expect this host to be the main source of *B. divergens* infections in ticks on farmland leading, presumably, to reduced genetic diversity.

While short (500 bp) 18S rRNA gene fragments that matched U16370 by 100% have been reported from red deer in Ireland [3], the question remains whether these isolates are indeed *B. divergens* or a different species that closely resembles *B. divergens* in the 18S rRNA locus, similar to *B. capreoli*. Even if they are identical with *B. divergens* it is unknown whether parasitaemias in red deer reach high enough levels to infect engorging ticks. It must also be borne in mind that, according to transmission studies that predate the development of PCR, infections acquired by tick larvae transovarially can be maintained to the second generation larvae even if the intervening tick stages feed on non-susceptible nonbovine host species [16]. If these observations hold true finding *B. divergens*-infected ticks at some distance from bovine hosts should come as no surprise.

## 4. Materials and Methods

### 4.1. Screening of Tick Isolates Using TaqMan PCR

Ticks analysed in this study were collected by blanket dragging and morphologically identified as described in [5]. Sampling sites included farmland, woodland, bog and limestone pavement habitats typical of the Burren region in the West of Ireland (Table 1). DNA was extracted from individual nymphs using the QIAGEN QIAamp^®^ DNA Mini Kit following homogenisation with stainless steel beads (Precellys^®^ Montigny-le Bretonneux, France) and overnight incubation in proteinase K (provided with the kit) [5]. Overall, 1369 ticks were screened for the presence of *B. divergens* using a TaqMan PCR protocol targeted at the heat shock protein 70 (hsp70) locus [15] (Table 2). All samples were tested in replicate in a total reaction volume of 20 μL containing 1.2 μL each of the forward and reverse primers (5 μM), 0.4 μL of the respective probes (5 μM), 10 μL of 2× FastStart Universal Probe Master (ROX) (Roche Diagnostics GmbH, Mannheim, Germany), 2.2 μL of nuclease-free water and 5 μL template DNA. Positive controls consisted of DNA extracted from *B. divergens* in vitro cultures, while nuclease-free water was used as negative control in all assays. Tick species identity was confirmed using a TaqMan PCR targeted at the internal spacer region 2 (ITS2) of *I. ricinus* as described by [5].

### 4.2. Confirmation of Positive Samples and Genotyping Using Nested Conventional PCR

Of the 1369 nymphs that were screened by TaqMan PCR, 27 samples resulted in a Ct value ≤ 40 in one or both of the replicates. In order to ensure no positives were missed, all 27 samples were subjected to nested PCR analysis aimed at the 18S rRNA gene [3] (Table 2). The PCR reaction mix consisted of 1x GoTaq Flexi PCR buffer, 1.5 mM MgCl_2_, 0.2 mM of each deoxynucleotide triphosphate, 500 nM of the forward and reverse primers, 1.25 U Promega GoTaq Flexi DNA Polymerase (Promega, Madison, WI, USA) and 5 µL DNA template (1st PCR) or 2 µL primary PCR product (in the nested PCR) in a total reaction volume of 50 µL was used. Negative controls (master mix with nuclease-free water instead of DNA template) were included in each assay.

All nested PCR amplicons were purified using the QIAquick PCR purification kit (Qiagen, Hilden, Germany) and sequenced in both directions using the internal PCR primers (Eurofins Genomics, Ebersberg, Germany). Consensus DNA sequences derived by aligning forward and reverse sequences were compared against published sequences using Clustal Omega. Tick infection rates in different sites were statistically compared based on percentages ± 95% confidence intervals.

## 5. Conclusions

In conclusion, we reported similar *B. divergens* infection rates in ticks collected from woodland and farmland. Moreover isolates from woodland ticks matched published sequences from cattle and humans more closely than did those from farmland. Since nymphs tend to quest close to where the larval stage has dropped off the previous host, our results indicate the potential presence of competent reservoir hosts for *B. divergens* in Irish woodlands. Considering that parasites that were 100% identical to *B. divergens* in two fragments of the 18S rRNA gene have been described from Irish red deer, they are the most likely candidates. In vitro isolation, followed by molecular characterisation and viability assays in human and bovine red blood cells similar to studies carried out by Malandrin and colleagues [11] would go some way to investigate the potential role of red deer in the epidemiology of *B. divergens*. However, bovine-cervine cross-infection trials will be necessary to resolve the question unequivocally.

## Figures and Tables

**Figure 1 pathogens-11-00312-f001:**
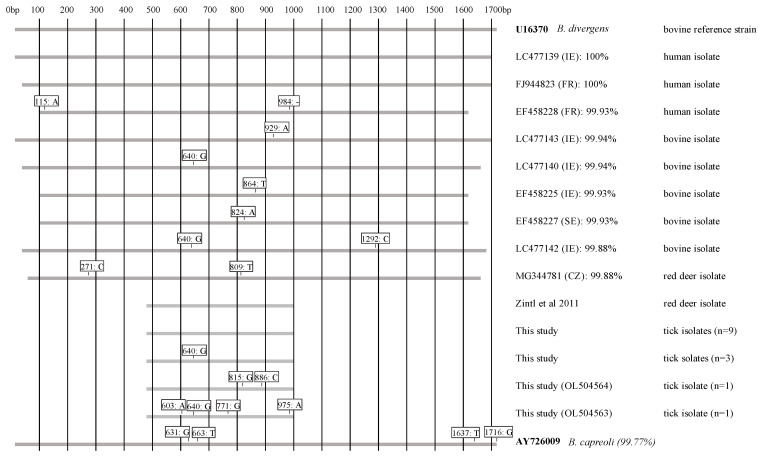
*Babesia divergens* 18S rRNA gene sequences isolated from ticks during the present study compared to corresponding sequences from *B. divergens* isolates from humans, cattle and red deer reported from Ireland and several other European countries (as indicated by country abbreviations in the legend). U16370 and AY726009 are treated as reference sequences for *B. divergens* and *Babesia capreoli* respectively. All numbers refer to positions in U16370. Identity scores also refer to U16370 and were derived using Clustal Omega. SNPs in positions 631, 663 and 1637 are considered characteristic for *B. capreoli* [11]. ‘-’ indicates a missing base.

**Table 1 pathogens-11-00312-t001:** *Ixodes ricinus* nymphs screened for the presence of *B. divergens.*

Habitat (Number of Sites)	Screened for *B. divergens*	Positive by Nested PCR (18S rRNA Gene) (% Positive ± 95% CI)
Woodland (*n* = 10)	633	10 (1.6% ± 1.0%)
Bogland (*n* = 1)	49	2 (4.1% ± 5.5%)
Limestone pavement (Burren) (*n* = 1)	50	0 (0%)
**Total**	**1369**	**14 (1.0% ± 0.5%)**

**Table 2 pathogens-11-00312-t002:** PCR target genes, primers, probes and protocols.

Gene Target (Length)	Primer and Probe Sequences	PCR Protocol
**TaqMan PCR** **Protocol** [15]		
hsp70 (83 bp)	Bab_di_hsp70_F: 5′CTCATTGGTGACGCCGCTA Bab_di_hsp70_R: 5′CTCCTCCCGATAAGCCTCTT Bab_di_hsp70_P:FAM-AGAACCAGGAGGCCCGTAACCCAGA-BHQ1	95 °C: 10 min 40 cycles: 95 °C: 15 s, 60 °C: 1 min
**Nested PCR** **Protocol** [3]		
18S rRNA gene (561 bp)	1st PCR: BTH-1F (F1): 5′ CCTGAGAAACGGCTACCACATCT BTH-1R (R1): 5′ TTGCGACCATACTCCCCCCA 2nd PCR: GF2 (F2): 5′ GTCTTGTAATTGGAATGATGG GR2 (R2): 5′ CCAAAGACTTTGATTTCTCTC	1st PCR: 94 °C: 10 min 40 cycles: 95 °C: 30 s, 57 °C: 40 s 72 °C: 1 min 72 °C: 10 min 2nd PCR: 94 °C: 10 min 40 cycles: 95 °C: 30 s, 50 °C: 40 s 72 °C: 1 min 72 °C: 10 min

## Data Availability

The datasets generated during and/or analyzed during the current study can be find within the main text.

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
