# Peer review of "The Prevalence and Genetic Diversity of Babesia divergens in Ixodes ricinus Nymphs Collected from Farm- and Woodland Sites in Ireland"

_pathogens, 2022, doi:10.3390/pathogens11030312_

Round 1
Reviewer 1 Report
The authors investigated prevalence of Babesia divergens in Ixodes ricinus collected from lands in Ireland. The authors did a lot of work and some results are interesting, however, due to the following reasons, I do not recommend this manuscript to be published in this journal.
- In abstract and results sections, without statistical analysis, terms like ‘comparatively high’, ‘slightly higher’ cannot be used.
- Based on the purpose of the study, the authors aimed to investigate the potential role of deer in the transmission cycle of B. divergens in Ireland, however, there is no evidence that the ticks collected came from deer.
- I recommend including map on the studied regions.
- divergens’ sequences identified in this study should be uploaded in public site. e.g. GenBank database.
- Table 2 were duplicated.
- How did the authors determine the tick species?
Author Response
Dear reviewer,
many thanks for the thorough review of our manuscript entitled ‘The Prevalence and Genetic Diversity of Babesia divergens in Ixodes ricinus Nymphs Collected from Farm- and Woodland Sites in Ireland’. In response to the comments and queries the following changes have been made:
- In abstract and results sections, without statistical analysis, terms like ‘comparatively high’, ‘slightly higher’ cannot be used.
Response: infection rates in ticks collected from farmland and woodland were compared using percentages ± 95% confidence intervals (lines 170-171), i.e. overlapping intervals indicate that there was no statistical difference. This has been corrected in the text (lines 27-28; 110; 181-182).
- Based on the purpose of the study, the authors aimed to investigate the potential role of deer in the transmission cycle of B. divergens in Ireland, however, there is no evidence that the ticks collected came from deer.
Response: That is correct. However, the point is that woodland nymphs didn’t have lower infection rates than ticks collected from farmland, which is what would have been expected in the absence of competent reservoir hosts. This has been clarified: ‘Considering that engorged ticks do not move very far and probably quest close to where they have dropped off [12], it is likely that the nymphs collected in woodland habitats had fed on wildlife hosts as larvae. Moreover, a significant proportion of them probably fed on deer, as these are the preferred hosts of all tick life cycle stages [13]. Interestingly, the B. divergensinfection rates were similar in nymphs collected from woodland and farmland sites suggesting that woodland nymphs may have become infected by engorging on deer and indicating a potential role of (red) deer as reservoir hosts for B. divergens.’ (lines 106-112)
- I recommend including map on the studied regions.
Response: The sample locations have already been published in Zintl et al 2020. This has been clarified (lines 140-142)
- divergens’ sequences identified in this study should be uploaded in public site. e.g. GenBank database.
Response: Novel sequences were logged in GenBank under accession numbers OL504563 and OL504564 (lines 91-93), for all other sequences the accession numbers for identical GenBank records have been provided: ‘The ~560bp 18S rRNA gene fragments of 9 isolates derived from ticks collected in woodland were 100% identical to U16370 which is widely used as a reference sequence for 18S rRNA gene fragments of B. divergens. Three further isolates, including one from woodland and two isolated from ticks collected in bogland, differed from U16370 in a single base in position 640 but matched two published Irish bovine isolates (LC477140 and LC477142) in this position.’ (lines 88-93)
- Table 2 were duplicated.
Response: apologies, this has been corrected
- How did the authors determine the tick species?
Response: the ticks had been identified morphologically and confirmed by TaqMan PCR as described in a previous paper (Zintl et al 2020). This has been clarified (lines 161-162)
Your sincerely,
Annetta Zintl
Reviewer 2 Report
Please see the attached annotated manuscript for detailed comments and queries.

Author Response
Dear reviewer,
many thanks for the thorough review of our manuscript entitled ‘The Prevalence and Genetic Diversity of Babesia divergens in Ixodes ricinus Nymphs Collected from Farm- and Woodland Sites in Ireland’. In response to the comments and queries the following changes have been made:
- Lines 29-30: infection rates in ticks collected from farmland and woodland were compared using percentages ± 95% confidence intervals (lines 170-171), i.e. overlapping intervals indicate that there was no statistical This has been corrected in the text (lines 27-28; 110; 181-182).
- Lines 44, 79, 80, 87: The words Babesia and divergens have been italicised (now in lines 44, 82, 83, 90)
- Table 1: Apologies, this was an oversight. The reference to phagocytophilum has been removed from the title. However, the designations ‘dairy’ and ‘beef’ farms have been retained.
- Figure 1: The species names have been italicised
- Table 2: The PCR targets, primers and probes should be under Methods. The use of the Taqman protocol for hsp70 should also be in Methods.
Response: The gene targets and PCR protocols are referred to in the Methods (sections 4.1 and 4.2). The table was mistakenly inserted twice , including once in the wrong place on page 4. This has now been corrected.
- Line 113: is hsp70 sequence information not available for bovine B. divergens? Response: It is, in fact that’s what the TaqMan PCR protocol developed by Michelet and colleagues is based on. The issue is that we do not have hsp70 sequence information for any of the Babesias detected in deer except B. odocoilei. This sentence has been modified: ‘However, since the only deer Babesia hsp70 gene sequences currently available in the database are for Babesia odocoilei, the possibility that the PCR developed by Michelet and colleagues [15] also amplifies other, closely related, wildlife species cannot be excluded categorically (lines 119-123).
- Line 121: is the isolate location not important?
Response: We are not quite sure what the reviewer means by this question. Most of the isolates compared in this study were derived from Ireland, with a small number reported from elsewhere (as indicated by the country abbreviations in Figure 1). This has been highlighted in the legend of Figure 1: ‘Babesia divergens 18S rRNA gene sequences isolated from ticks during the present study compared to corresponding sequences from B. divergens isolates from humans, cattle and red deer reported from Ireland and several other European countries (as indicated by country abbreviations in the legend).’ (lines 97-103)
- Section 4.1: How were Ix. ricinus identified? Are there related Ix ticks in this part of Ireland?
Response: the ticks had been identified morphologically and confirmed using TaqMan PCR as described in a previous paper (Zintl et al 2020). This has been clarified (lines 161-162)
- Line 175: The statement is unjustifiably strong.
Response: The statement has been qualified as follows: ‘Since nymphs tend to quest close to where the larval stage has dropped off the previous host, our results indicate the potential presence of competent reservoir hosts for B. divergens in Irish woodlands. Considering that parasites that were 100% identical to B. divergens in two fragments of the 18S rRNA gene have been described from Irish red deer, they are the most likely candidates.’ (lines 183-188).
Yours sincerely, Annetta Zintl
Reviewer 3 Report
The authors explore the distribution of Babesia divergens in Ixodes ricinus ticks, collected from different environments in Ireland. I only have some minor comments regarding the manuscript, which the authors can find below.
First of all, the last sentence in the abstract states that the results found by the authors support the idea that red deer may have an important role in the transmission cycle of B. divergens. In my opinion, there is no evidence for this in the paper. The authors did not perform blood meal analysis (which would suggest that the infected ticks fed on red deer), or the infected ticks weren’t collected from deer. Therefore, I do not see how the authors drew this conclusion in their work. This statement appears again in the paper later, which is still not justified for me. I suggest removing this.
P1, L42: I think it would be interesting to add a yearly average number of this infection in humans, if known.
P2, L61: again, if there is a known number of infection cases in humans please indicate.
P2, L64: there is a really sudden change in topic here, without connection, going from human infection numbers right to deer faunistic data in Ireland. I suggest adding one or two sentences to connect the two thoughts.
Figure 1: legends are too small, hard to read, I suggest modifying it.
P4, L107-109: strange sentence. Also, it just further supports that just because infected ticks were found in woodlands, does not necessarily mean that they were fed on deer (the conclusion of the authors, that deer are important reservoirs).
P4, L114 AND P5, L130: so how did the authors conclude that it is indeed B. divergens that they found, if the PCR is not specific enough? Please explain or correct in text accordingly. If, they cannot surely conclude that it is B. divergens then they cannot report it In the result as B. divergens, but Babesia sp.
P6, L157: In Table 1 it says 14 ticks were infected, what does this 27 positive individual refers to?
P6, l157: again, these results, does not indicate this conclusion. Please remove.
Author Response
Dear Reviewer
many thanks for the thorough review of our manuscript entitled ‘The Prevalence and Genetic Diversity of Babesia divergens in Ixodes ricinus Nymphs Collected from Farm- and Woodland Sites in Ireland’. In response to the comments and queries the following changes have been made:
- First of all, the last sentence in the abstract states that the results found by the authors support the idea that red deer may have an important role in the transmission cycle of B. divergens. In my opinion, there is no evidence for this in the paper. The authors did not perform blood meal analysis (which would suggest that the infected ticks fed on red deer), or the infected ticks weren’t collected from deer. Therefore, I do not see how the authors drew this conclusion in their work. This statement appears again in the paper later, which is still not justified for me. I suggest removing this.
Response: It is correct that no blood meal analysis was carried out, nor were ticks collected directly from deer. It could be argued that neither of these approaches would have provided unequivocal proof that the protozoa detected in the ticks originated from deer either . However, we do concede the point that our evidence is weak. Therefore the last sentence in the abstract has been deleted as suggested. In the discussion our reasoning was explained further as follows: ‘Considering that engorged ticks do not move very far and probably quest close to where they have dropped off [12], it is likely that the nymphs collected in woodland habitats had fed on wildlife hosts as larvae. Moreover, a significant proportion of them probably fed on deer, as these are the preferred hosts of all tick life cycle stages [13]. Interestingly, the B. divergens infection rates were similar in nymphs collected from woodland and farmland sites suggesting that woodland nymphs may have become infected by engorging on deer and indicating a potential role of (red) deer as reservoir hosts for B. divergens.’ (lines 106-112).
- P1, L42: I think it would be interesting to add a yearly average number of this infection in humans, if known.
Response: The information has been provided (lines 39-40)
- P2, L61: again, if there is a known number of infection cases in humans please indicate.
Response: The number of reported zoonotic infections in Ireland has been included (lines 60-62)
- P2, L64: there is a really sudden change in topic here, without connection, going from human infection numbers right to deer faunistic data in Ireland. I suggest adding one or two sentences to connect the two thoughts.
Response: This has been done: ‘Regarding Babesia infections in Irish deer, previous work has identified isolates from red deer that were 100% identical to bovine B. divergens sequences in two fragments of the 18SrRNA gene (Zintl et al 2011) (lines 64-66)
- Figure 1: legends are too small, hard to read, I suggest modifying it.
Response: The font size of the legend and inscriptions in Figure 1 has been increased. If the figure were to be reproduced in landscape format, the font size could be increased further.
- P4, L107-109: strange sentence. Also, it just further supports that just because infected ticks were found in woodlands, does not necessarily mean that they were fed on deer (the conclusion of the authors, that deer are important reservoirs).
Response: This paragraph has been modified. Please see response to 1)
- P4, L114 AND P5, L130: so how did the authors conclude that it is indeed B. divergens that they found, if the PCR is not specific enough? Please explain or correct in text accordingly. If, they cannot surely conclude that it is B. divergens then they cannot report it In the result as B. divergens, but Babesia sp.
Response: The protozoans were identified based on 2 genetic loci, the hsp70 and 18S rRNA gene, which is generally considered sufficient for unequivocal species identification. The point being made in the discussion is that in the absence of HSP70 sequence information from ‘B. divergens-like’ isolates from red deer we can’t categorically exclude the possibility of non-specific amplification. However, it is extremely unlikely. This has been clarified in the text: ‘It is reasonable to assume that the same is true for other Babesia species. However, since the only deer Babesia hsp70 gene sequences currently available in the database are for Babesia odocoilei, the possibility that the PCR developed by Michelet and colleagues [15] also amplifies other, closely related, wildlife species cannot be excluded categorically. Nevertheless it is important to note that the primers and probes used in the protocol do not match the hsp70 gene of B. odocoilei.’ (lines 118-124)
- P6, L157: In Table 1 it says 14 ticks were infected, what does this 27 positive individual refers to?
Response: Of the 1369 nymphs that were screened by TaqMan PCR, 27 samples resulted in a Ct value ≤ 40 in one or both of the replicates. In order to ensure no positives were missed, all 27 samples were subjected to nested PCR aimed at the 18S rRNA gene. This has been clarified in the text (lines 165-167)
- P6, l157: again, these results, does not indicate this conclusion. Please remove.
Response: This part of the conclusions has been rephrased as follows: ‘Since nymphs tend to quest close to where the larval stage has dropped off the previous host, our results indicate the potential presence of competent reservoir hosts for B. divergens in Irish woodlands. Considering that parasites that were 100% identical to B. divergens in two fragments of the 18S rRNA gene have been described from Irish red deer, they are the most likely candidates.’ (lines 183-188)
Yours sincerely, Annetta Zintl
Round 2
Reviewer 1 Report
At least, the authors should declare that the ticks were not directly collected from deer as a limitation of the study.
Author Response
Reviewer 1:
single comment following the 2nd round
At least, the authors should declare that the ticks were not directly collected from deer as a limitation of the study.
Response: In our experience, results from ticks collected directly from deer are singularly difficult to interpret, as any pathogens they carry may originate from the current or former blood meal or indeed from nearby ticks acquired during co-feeding. Of course we agree that there are important limitations to the study, but the best way to address them would be by in vitro isolation of Babesias from red deer, more detailed molecular characterisation and bovine-cervine cross-infection trials. This is stated in lines 188 to 192.